# Consistent survival in consecutive cases of life-supporting porcine kidney xenotransplantation using 10GE source pigs

Daniel Eisenson [1], Yu Hisadome[1], Michelle Santillan [1], Hayato Iwase[1], WeiLi Chen[1,2], Akira Shimizu [2], Alex Schulick[1], Du Gu[1], Armaan Akbar[1], Alice Zhou[1], Kristy Koenig[1], Kasinath Kuravi[3], Farzana Rahman[3], Lori Sorrells[3], Lars Burdorf[3], Kristina DeSmet[3], Daniel Warren[1], Leigh Peterson[3], Marc Lorber[3], David Ayares[3], Andrew Cameron[1] & Kazuhiko Yamada [1] ✉

Xenotransplantation represents a possible solution to the organ shortage crisis and is an imminent clinical reality with long-term xenograft survival in pig-to-nonhuman primate (NHP) heart and kidney large animal models, and short-term success in recent human decedent and clinical studies. However, concerns remain about safe clinical translation of these results, given the inconsistency in published survival as well as key differences between pre-clinical procurement and immunosuppression and clinical standards-of-care. Notably, no studies of solid organ pig-to-NHP transplantation have achieved xenograft survival longer than one month without CD40/CD154 costimulatory blockade, which is not currently an FDA-approved immunosuppression strategy. We now present consistent survival in consecutive cases of pig-to-NHP kidney xenotransplantation, including long-term survival after >3 hours of xenograft cold preservation time as well as long-term survival using FDA-approved immunosuppression. These data provide critical supporting evidence for the safety and feasibility of clinical kidney xenotransplantation. Moreover, long-term survival without CD40/CD154 costimulatory blockade may provide important insights for immunosuppression regimens to be considered for first-in-human clinical trials.

Xenotransplantation represents a promising solution to the donor organ shortage. The next steps toward broader clinical translation have been outlined by the Food and Drug Administration (FDA), which recently issued guidance on a regulatory path forward for solid organ xenotransplantation[1]. The regulatory framework provided by the FDA has brought increased scrutiny to the field with respect to the consistency of results, the method of organ procurement and preservation, and the use of non-FDA-approved medications in immunosuppression regimens.

Although the FDA has not yet approved clinical trials in kidney xenotransplantation, demonstration of consistent >6 months survival in consecutive preclinical NHP experiments may be required[2]. While there have been multiple examples of long-term survival in NHP studies, no published case series has shown consistent (>50%) survival beyond 6 months in consecutive (>3 recipients) experiments. In most studies, recipients who experience hyperacute xenograft rejection or early adverse outcomes due to technical complications are excluded[3,4]. However, transparent demonstration of consecutive long-term

[1]Department of Surgery, Division of Transplantation, The Johns Hopkins School of Medicine, Baltimore, MD, USA. [2]Department of Pathology, Nippon Medical School, Tokyo, Japan. [3]United Therapeutics Corporation, Silver Spring, MD, USA. ✉e-mail: kyamada6@jhmi.edu

survival, without exclusion of adverse outcomes, is critical to determining safety and efficacy as we consider xenografts in humans.

Additionally, no published studies have demonstrated long-term porcine kidney xenograft survival after significant cold ischemic time. This raises potential concerns about the feasibility of off-site procurement at a designated pathogen-free (DPF) facility, which will likely be required for clinical trials, as cold ischemic time will be incurred for the transport of the xenograft. While cold ischemic time is well-tolerated in allogeneic kidney transplantation, our recent study has demonstrated that standard-of-care static cold storage of five hours leads to hyperacute graft loss across xenogeneic barriers in pig-to-NHP kidney transplantation[5].

Finally, the prospect of clinical trials has also renewed debate about whether non-FDA-approved medications, particularly CD40/CD154 costimulatory blocking agents, are necessary. To date, no study has demonstrated long-term solid organ xenograft survival without CD40/CD154 blockade[3]. On the other hand, clinicians have made remarkable progress preventing rejection of allografts transplanted across ABO / HLA incompatible barriers using calcineurin inhibitor (CNI) based immunosuppression regimen without CD40/CD154 costimulation blockade. The gulf between effective immunosuppression regimens for human allotransplantation and NHP xenotransplantation experiences raises concerns for the clinical translation of preclinical NHP data.

In this study, we present the first demonstration of consistent survival in consecutive cases of pig-to-NHP kidney xenograft transplantation. Moreover, we demonstrate the feasibility of a regulatory-compliant, porcine kidney procurement, and preservation strategy, showing that porcine kidneys procured at a DPF facility can be preserved adequately and transplanted successfully after 3–5 hours of cold preservation time. Perhaps most importantly, we also bridge this decades-long gap between NHP and clinical transplantation immunosuppressive strategies, with the first demonstration of long-term xenograft survival in genetically modified pig-to-baboon kidney transplantation using conventional, clinically available immunosuppression. We believe these data, while preliminary, may serve as proof of concept that CNI-based immunosuppression regimens can effectively prevent xenograft rejection in carefully selected recipients of genetically modified porcine kidneys.

## Results

This study is the first report of pig-to-NHP kidney xenotransplantation experience using source pigs with 10 genetic modifications (referred to as 10GE pigs); it is part of a larger Investigational New Drug (IND)-enabling nonclinical study supported by United Therapeutics. All cases were performed consecutively during the first and second quarter of 2023. No cases were excluded. Each baboon recipient received one porcine kidney, transplanted orthotopically with a porcine renal vein and artery anastomosed to the inferior vena cava and the aorta, respectively. Both native baboon kidneys were removed at the time of transplantation, and the xenografts were immediately life-supporting, producing urine on the operating table. One recipient (Group 2, CD40/CD154 costimulatory blockade maintenance immunosuppression) with high levels of preformed anti-source pig Nabs experienced hyperacute xenograft rejection, as retrospectively explained by modified immunologic screening methodology (designated as a high-risk recipient by elevated IgG binding). All recipients with low levels of preformed anti-source pig Nabs accepted their grafts and maintained stable graft function for 337, 285, 278, 252, and 165 days (Figs. 1a and 1b, details in figure legend). Among all recipients, mean and median survival was 220 days and 261 days, respectively; among five low-risk recipients, mean survival was 263 days (±49.5 [CI: 213–313 days]), and the median was 278 days. Four of five low-risk recipients maintained graft function beyond six months (histology of xenograft biopsies at this timepoint shown in Fig. 2); three of the remaining four recipients

maintained graft function beyond nine months. One recipient (B3520, Group 2, CD40/CD154 costimulatory blockade maintenance immunosuppression) developed antibody-mediated rejection at the time of necropsy on POD278 (Fig. 3), with IgG and IgM antibody-binding as well as a slight increase in circulating donor-specific antibodies (DSA) (See supplementary information). Another recipient (B0820, Group 1, CNI-based immunosuppression) developed progressive renal failure. which met endpoint criteria on POD285 and was found on necropsy to have histologic evidence of acute cellular xenograft rejection with lymphocyte infiltrate. The other three recipients (B9819, B4519, and B1320, all Group 2, CD40/CD154 costimulatory blockade maintenance immunosuppression) lost their grafts suddenly on POD165, POD252, and POD337, respectively. In each case, rapid decline in graft function was preceded by symptoms of upper respiratory tract infection and adenovirus seroconversion consistent with new infection. Necropsy kidney specimens stained positive for adenovirus (Fig. 4). One of these recipients (B4519, necropsy histology included in Fig. 3) also had weak IgG and IgM binding in the xenograft (Fig. 4), although elicited DSA was negative (see supplementary information). Among the remaining two recipients, there were no elicited DSAs, and only minimal and nonspecific IgG/IgM binding assessed by immunofluorescence of necropsy xenograft specimens (Fig. 4).

Xenograft kidneys maintained normal concentrations of sodium, potassium, and chloride until the development of irreversible renal failure. B9819 developed dehydration around POD100 associated with hyponatremia and hypochloremia, which normalized with fluid resuscitation. Mild-to-moderate hypercalcemia was observed in long-term survivors (see supplementary information), a physiologic change that has been previously reported after porcine renal xenograft transplantation in a pig-to-macaque model[6]. Kidney graft volume remained stable throughout postoperative course, in contrast with previous studies using GalTKO source pigs[7].

## Discussion

There has been enormous progress in the field of xenotransplantation over the last twenty years, with recent reports of life-supporting pig-to-NHP heart and kidney xenografts enabling survivals of >6 months. However, there are three major concerns with these data as we approach clinical trials in solid organ xenotransplantation: 1) reports of long-term survivals have been restricted to a small number of recipients, with nearly half of recipients in recent landmark studies rejecting their grafts within the first month after transplantation[4,8]; 2) procurements and ischemic times are not reflective of the clinical and regulatory realities that will govern clinical trials; and 3) CD40/CD154 costimulatory blockade, which is not currently FDA-approved, has been used in each of these cases. This study is the first of its kind to report durable 220-day (mean) and 261-day (median) survival in a series of six consecutive NHP xenotransplants, procured at a DPF facility and transported with FDA-approved preservation technologies and includes the first demonstration of long-term solid organ xenograft survival without CD40/CD154 blocking agents.

There are a number of reasons for the small sample size in most studies, including the high cost of pig-to-NHP experiments and the limited availability of genetically modified source pigs and NHPs. More concerning is the relatively high rate of early graft loss due to hyperacute rejection, accelerated antibody-mediated rejection, or early death from infection. Hyperacute rejection, which once plagued early attempts at pig-to-NHP transplantation in the era before genetically modified source pigs, still impacts early graft survival; yet, there are no established criteria to guide recipient selection. In a recent published report, Anand and colleagues attribute variable survival to differences between NHPs and humans, as porcine genetic modifications

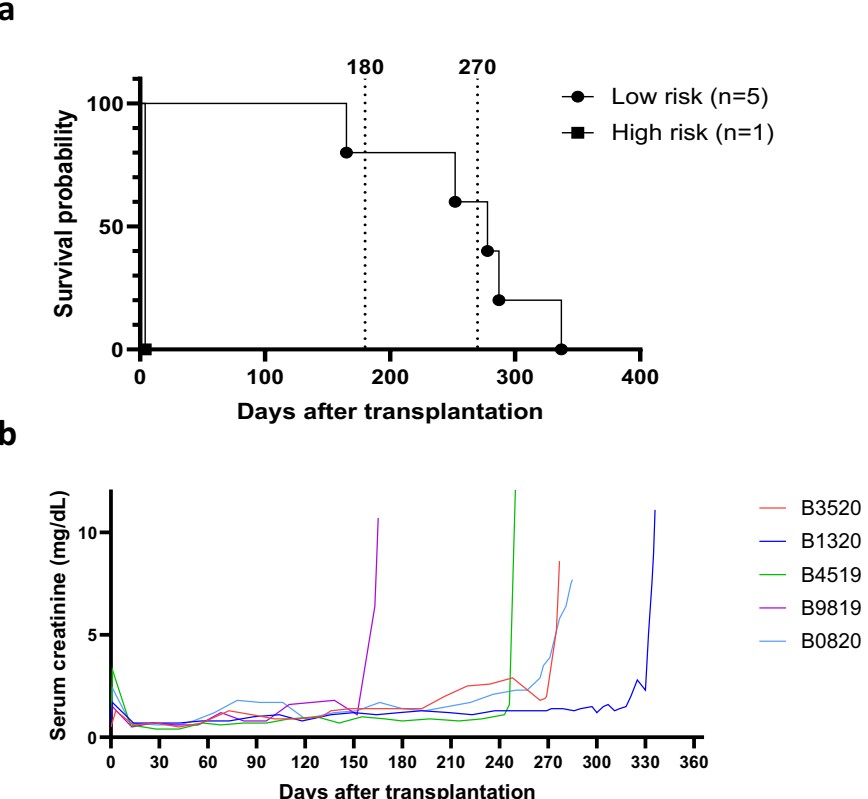

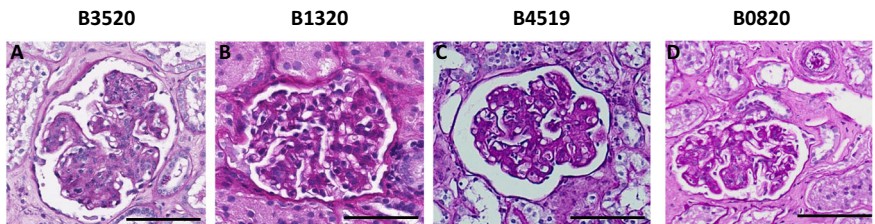

**Fig. 1 | 10GE porcine kidneys support life with stable renal function in baboons.**
**a** Survival probability for recipients transplanted with 10GE renal xenografts. One
recipient with high levels of preformed antibody to source pig 10GE cells (denoted
as "High risk") hyperacutely rejected graft. The remaining five recipients (denoted
as "Low risk" for hyperacute graft loss) lost their grafts on POD165, POD252,
POD285, and POD337. **b** Serum creatinine levels among these five recipients
remained within the normal range until the development of irreversible graft
failure.

**Fig. 2 | Four of five xenograft recipients maintained stable graft function
beyond six months.** Periodic acid-Schiff staining of xenograft kidney biopsy
sample derived from recipients B3520 (**A**), B1320 (**B**), B4519 (**C**), and B0820 (**D**) at
the time of scheduled six-month (POD180) biopsy. Note: biopsy from B3520 (**A**)
was taken on POD214 due to inadequate POD180 specimen. Multiple cuts of each
sample were obtained and evaluated with similar findings. The black bar at the
bottom right of each panel corresponds to 100 μm.

(particularly *CMAH* inactivation) have been designed for human
recipients[8]. While we agree that disparities between recipient species
may account for variable immunologic responses to porcine xeno-
grafts, we believe that it is critical to validate predictive immunologic
screening strategies in the preclinical setting prior to clinical
translation.

   In this study, we carefully selected recipients to avoid hyperacute
graft loss according to screening methodology developed by The
Yamada Lab at Johns Hopkins University[9], which has subsequently
been informed and modified in response to our results (see supple-
mentary information). Although we initially used triple knock-out
(TKO) peripheral blood mononuclear cells (PBMCs) as donor-
surrogates for immunologic testing, we retrospectively found that
source-pig specific testing is critical: the single recipient who experi-
enced hyperacute xenograft rejection had high levels of preformed
anti-10GE IgG antibody. These results illustrate the importance of

source-pig-specific screening, and we anticipate this will be critical to
the validation of preformed antibody assessments used to select
patients for clinical trials in xenotransplantation. In contrast to the
recent study by Anand and colleagues, where over 40% of grafts were
lost within the first month after transplantation, all recipients identi-
fied as low-risk based on pre-transplant screening in this study
accepted their grafts and maintained stable graft function throughout
the early postoperative period.

   The consistency of our results may also be attributed to reliable
transgene expression in the 10GE source pigs. Whereas in previous
xenokidney transplants from multi-transgenic source pigs[10], trans-
genic expression was achieved by insertion of a multi-copy, randomly
integrated vector under control of a specific promoter, here, transgene
insertion was targeted to the α1,3GT locus and the CMAH locus, and all
six transgenes were combined on only two multi-cistronic vectors
(Fig. 5). Targeted insertion of transgenes using these multi-cistronic

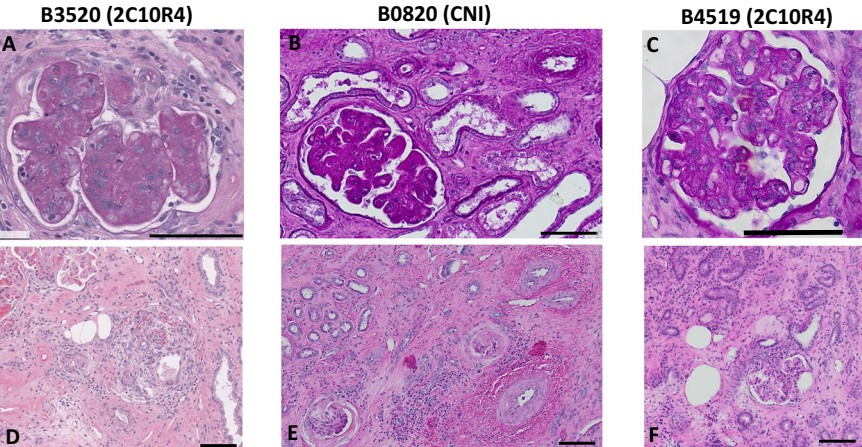

**Fig. 3 | Three of five xenograft recipients with histologic evidence of rejection at time of necropsy.** Top panels are periodic acid-Schiff staining of xenograft kidney biopsy sample derived from recipients B3520 (**A**), B0820 (**B**), and B4519 (**C**) at the time of necropsy; Hematoxylin and eosin (H&E) staining of B3520 (**D**), B0820 (**E**), and B4519 (**F**) at time of necropsy. Multiple cuts of each sample were obtained and evaluated with similar findings. The black bar at the bottom right of each panel corresponds to 100 μm.

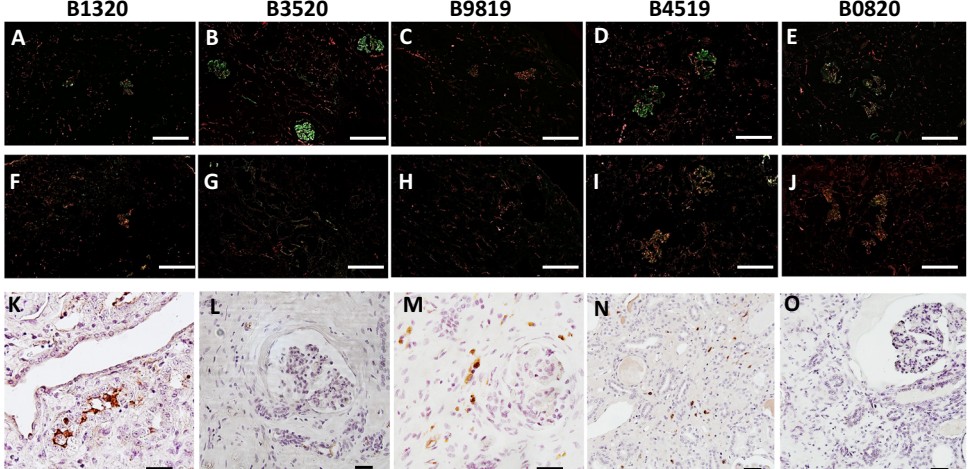

**Fig. 4 | Xenograft loss in remaining animals associated with adenovirus infection in the absence of antibody binding in xenograft.** Immunofluorescence staining of IgM (upper panels, **A**–**E**) and IgG (middle panels, **F**–**J**) of 10GE xenograft biopsies obtained from recipients at time of necropsy, double-stained with CD31 and performed in duplicate to confirm findings. White bar at the bottom right of each (**A**–**J**) corresponds to 200 μm. Immunohistochemistry staining for human adenovirus (lower panels, **K**–**O**); staining performed in duplicate to confirm findings. Black bar at bottom right of each (**K**–**O**) corresponds to 40 μm.

vectors has further benefits including co-segregation of the transgenes to offspring during breeding, as well as fewer off-target effects.

While three to five months have regularly been used as pre-determined endpoints in large animal models of transplantation[11–14], due in part to the challenges inherent to the complex medical management of large animals, survivals have been open-ended in xenotransplantation to allow for greater follow-up. All five low-risk recipients in this study survived beyond the five-month threshold that has recently been used to support the translation of experimental immunosuppressive reagents from preclinical to clinical trials in transplantation[15–17]. With enhanced follow-up in this study, we found that two recipients ultimately rejected their grafts: one recipient (B3520) on POD278 with evidence of AMR and another recipient (B0820) on POD285 with evidence of acute cellular rejection. Three recipients lost grafts suddenly after symptoms of upper respiratory tract infection and were found on necropsy to have histologic evidence of adenovirus nephropathy. Although the primary focus of zoonosis in xenotransplantation has been on the potential transmission of infection from xenograft to host, zoonosis is bi-directional, and transmission of host infections (including human adenovirus, which has long been understood to infect pig cells[18,19]) may also contribute to early graft loss.

Effective xenograft procurement and tolerance of cold preservation time are also critically important for the success of clinical trials. FDA regulations mandate that procurements must occur in a DPF animal facility, and this will require a period of ischemic time as these porcine organs are transported to the recipient transplant center. To our knowledge, no prior studies have described long-term xenograft survival after prolonged (>3 hours) cold preservation time; in fact, we have previously shown that five hours of static cold storage may precipitate hyperacute graft loss in pig-to-NHP kidney xenotransplantation[5]. In light of these results, we used two FDA-approved devices to evaluate hypothermic machine perfusion during transportation. We are actively investigating the mechanisms of ischemic reperfusion injury associated with static cold storage, and the mechanisms underlying the apparent protective effect of hypothermic machine perfusion.

Importantly, this case series includes the first demonstration of long-term solid organ xenograft survival achieved without CD40/CD154 costimulatory blockade. While the essential role of costimulation blockade in xenotransplantation is often attributed to unique

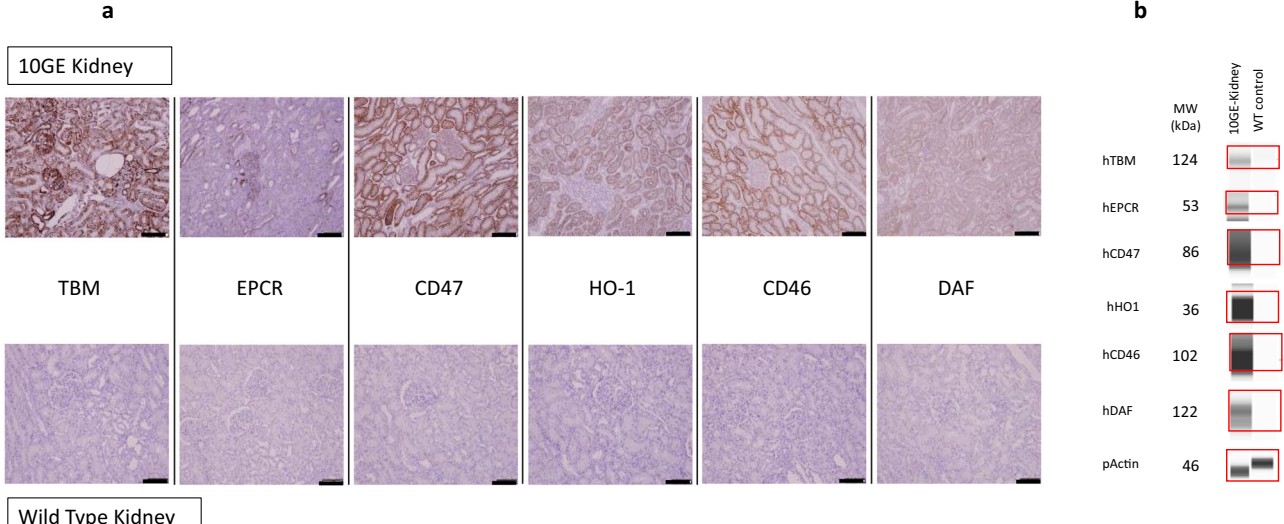

**Fig. 5 | Expression of human transgenes in 10GE porcine kidneys. A** Transgenic expression of hTBM, hEPCR, hCD47, hHO1, hCD46, and hDAF in kidney tissue as assessed by immunohistochemistry. Top panel is 10GE kidney tissue; bottom panel is wild-type kidney tissue (control). IHC staining was performed twice on each specific kidney sample. Black bar at the bottom right of each panel corresponds to 100 μm. **B** hTBM, hEPCR, hCD47, hHO1, hCD46 and hDAF expressed at expected molecular weight in 10GE kidney sample. WT tissue lysate acts as negative control. Actin acts as loading control for the assay. Representative data from two independent runs.

immunologic hurdles associated with interspecies transplantation, the CD40/CD154 immunosuppression paradigm may compound the uncertainty associated with the clinical translation of xenotransplantation. Although multiple drugs targeting this pathway are in development with promising early results, no agents have yet demonstrated non-inferiority in clinical trials, and no drugs have yet obtained FDA approval for use in transplantation. This appears to present a difficult choice for regulators, clinicians, and patients: is it safer to use a conventional immunosuppression regimen with a xenograft despite a lack of preclinical evidence in pig-to-NHP solid organ transplantation, or is it safer to use a clinically unproven immunosuppression regimen on the basis of preclinical pig-to-NHP data?

To date, three centers have performed pig-to-human solid organ transplantation, and each center has opted for a different immunosuppression strategy. New York University and The University of Alabama used conventional immunosuppression for their preclinical pig-to-decedent kidney transplants; whereas The University of Maryland included CD40/CD154 blockade, KPL-404 (Kiniksa Pharmaceuticals), which targets the 2C10 binding epitope on CD40, and tegoprubart (Eledon Pharmaceuticals), which targets CD154 (CD40L), for their two clinical pig-to-human heart transplants, respectively. Although early results have been encouraging at all three centers, no long-term (>180 days) survival data exists in pig-to-human transplantation to provide clarity. Since rejection often occurs beyond 60 days, it is unlikely that decedent studies with extended survivals can conclusively resolve this uncertainty.

This study provides the first evidence that conventional, FDA-approved immunosuppression regimens can be used to achieve long-term xenograft survival in pig-to-NHP solid organ transplantation. Indeed, 285-day xenograft survival using CNI-based immunosuppression with CD28/CTLA4 costimulation blockade is among the longest-reported xenograft survivals ever reported in life-supporting pig-to-baboon xenotransplantation. We believe that our implementation of this regimen, including careful monitoring of CNI levels and titration of CNI dose, as well as the use of CD28/CTLA4 costimulation blockade, are central to this recipient's long-term survival. While these data must be replicated, they help to resolve an important discrepancy between preclinical pig-to-NHP xenotransplantation data and the clinical allotransplantation experience; as such, they have immediate implications for imminent xenotransplantation clinical trial design. At present, these implications may be limited to recipients of 10GE xenografts, as reliable transgenic expression of multiple human complement regulatory, anti-coagulant, and anti-inflammatory transgenes in 10GE source pigs may obviate the need for CD40/CD154 blockade.

## Methods

### Genetically modified source pigs

The 10GE source pigs, whose genetic modifications are identical to those used in both of the pig-to-human cardiac clinical cases[20,21], were produced by somatic cell nuclear transfer of clonally derived fibroblasts, which were genetically modified with 10 gene edits, including the removal of three known carbohydrate antigen targets of preformed natural antibodies plus addition of human complement regulatory proteins and human anti-coagulant proteins (*Revivicor Inc*, Blacksburg, VA). Expression of human transgenes was confirmed by immunofluorescence and western blot (Fig. 5). To minimize off-target effects and ensure consistent transgene expression, all transgenes were inserted into only two loci: the α1,3GT locus, and the CMAH locus. A 2-gene multicistronic vector (including human transgenes hCD46 and hDAF) was integrated at the α1,3GT locus, and a 4-gene multicistronic vector (including human transgenes hTHBD, hEPCR, hCD47, and hHO1) was integrated at the CMAH locus, inactivating one allele at both the α1,3GT locus, and the CMAH locus. The second alleles of the GGTA1 and CMAH genes were inactivated by neoR insertion via homologous recombination and CRISPR/Cas9-mediated NHEJ. Both alleles of β4GalNT2 and GHr genes were inactivated by CRISPR/Cas9-mediated NHEJ. Preparation and procurement of porcine kidneys were performed at the Revivicor facility. Euthanasia was performed prior to organ procurement. Swine were maintained, treated, and euthanized according to guidelines established by the Animal Welfare Act and the NIH for the housing and care of laboratory animals and in compliance with protocols approved by both Revivicor and the Johns Hopkins University School of Medicine Institutional Animal Care and Use Committees.

### NHP recipients

The six baboon (*Papio anubis*) recipients were specific pathogen-free and provided by the Michale E. Keeling Center for Comparative

**Table 1 | Summary of recipient weights, source pig weights, preservation modality, and preservation time**

| Recipient ID | Recipient Sex | Recipient Weight (kg) | Pig Weight (kg) | Xenokidney Weight (g) | Preservation Modality | Preservation Time (hours) |
|---|---|---|---|---|---|---|
| B1320 | M | 9.5 | 17 | – | Lifeport | 3.5 hours |
| B3520 | M | 8.5 | 17 | – | XVIVO | 3.0 hours |
| B0820 | F | 10.5 | 24 | 56.4 | XVIVO | 5.2 hours |
| B4519 | F | 10 | 24 | 51.2 | Lifeport | 4.0 hours |
| B9819 | M | 10.5 | 18 | 38.6 | Lifeport | 3.7 hours |
| B10419 | F | 12.5 | 18 | 38.6 | Lifeport | 4.5 hours |

Medicine and Research (MD Anderson Cancer Center, Bastrop, TX). The baboons ranged from 2–4 years old and weighed between 8 to 12.5 kg. Equal numbers of male (3) and female (3) baboons were included. All baboons were maintained, treated, and euthanized at the study endpoint according to guidelines established by the Animal Welfare Act and the NIH for the housing and care of laboratory animals and in compliance with protocols approved by the Animal Care and Use Committee at The Johns Hopkins University School of Medicine.

### Animal husbandry
Animal husbandry was conducted in accordance with each facilities' SOPs. Swine were socially housed in individual pens on tenderfoot flooring in spaces with at least one and up to seven additional pigs. Swine were surveilled for pathogens of potential concern to NHP recipients. Baboons were housed both singly and in pairs in stainless-steel cages with access to an automatic watering valve. Temperature was maintained between 24 and 29 °C, humidity maintained between 30-70%, and lighting was provided on a light/dark cycle for approximately 12 hours each day. Baboons were fed certified chow with Lab-Diet 5038−Monkey Diet, with enrichment in the form of fruits and vegetables provided once daily. Water was available ad libitum throughout the studies for both swine and baboons.

### Pre-transplant screening
Prospective recipients were evaluated for anti-pig antibodies by complement-dependent cytotoxicity (CDC) using peripheral blood mononuclear cells (PBMCs) derived from triple antigen knockout source pigs. The presence of donor-specific antibodies was assessed by IgG and IgM antibody binding of baboon sera to 10GE-derived PBMCs. Initial screening was performed with donor-surrogate peripheral blood mononuclear cells (PBMCs) using triple knockout (TKO) pigs. Although B10419 had low TKO IgG antibody binding, B10419 was retrospectively found to have high anti-10GE (source pig) antibodies. Subsequent screening is performed using source-pig cells according to screening methodology developed by The Yamada Lab at Johns Hopkins University[9]. Recipients are selected for complement-dependent cytotoxicity (CDC) < 20% at 1:4 dilution, serum IgG binding on antibody flow cytometry less than the pretransplant value of a known rejector recipient, and 10GE/Gal knockout (GalTKO) IgG binding ratio <1.5.

### Induction and maintenance immunosuppresssion
All recipients underwent the same immunosuppression induction protocol: B cell depletion was performed with rituximab (20 mg/kg) on POD-6, T cell depletion on POD −3 and POD −1 (10 mg/kg and 5 mg/kg, rATG), and peri-transplant C1 esterase inhibitor (Berinert, 17.5 mg/kg) was used to prevent complement activation on POD −1, 0, and 1. Baboons then received either conventional immunosuppression (Group 1, one recipient) or CD40/CD154 costimulatory blockade-based maintenance immunosuppression (Group 2, five recipients). The conventional immunosuppression regimen featured only clinically available reagents: mycophenolate mofetil (MMF, 70 mg/kg/day) from postoperative day (POD) −5, tacrolimus (0.1-0.15 mg/kg/day) from POD −1, belatacept (CTLA4-Ig, 10 mg/kg) on POD 2 (continued weekly),

methylprednisolone initiated on POD 6 (2 mg/kg/day tapered to 0.5 mg/kg/day), and sirolimus initiated on POD 42 (0.1 mg/kg/day). The CD40/CD154 costimulation blockade-based regimen included the mouse-rhesus IgG4 chimeric anti-CD40 monoclonal antibody (2C10R4, targeting the 2C10 epitope of the CD40 molecule, dosed at 25 mg/kg twice weekly after loading dose) and anti-IL6R antibody (tocilizumab, 5 mg/kg biweekly), as well as MMF (70 mg/kg/day) and sirolimus (0.1 mg/kg/day), as in the conventional regimen.

### Porcine kidney transplantation
The porcine kidneys were procured from three 10GE source pigs, ranging from 17 to 24 kg in weight at the time of organ procurement, at Revivicor's DPF facility in Blacksburg, VA. Porcine kidneys were transplanted into three pairs of baboon recipients (six recipients total). The kidneys used in this study were preserved via oxygenated hypothermic machine perfusion (XVIVO, Kidney Assist Transport) or non-oxygenated hypothermic machine perfusion (Lifeport, Organ Recovery Systems) for between three and five hours of transportation time. Both perfusion devices used are FDA-approved devices for the preservation, transportation, and eventual transplantation into a recipient. Weights of source pigs, recipients, and xenografts, as well as hypothermic machine perfusion device used and preservation times are summarized in Table 1. Kidney failure as defined by creatinine >10 mg/dL was defined as the primary endpoint. All animals were used in compliance with guidelines provided by the Animal Care and Use Committee at The Johns Hopkins University School of Medicine.

### Histologic preparation and analysis
Kidney biopsy specimens were either a) frozen or b) fixed in 10% formaldehyde and embedded in paraffin. Frozen samples were used for immunofluorescence (IF). Anti-human IgG, IgM, and C3 (DAKO, Carpentaria, CA) all conjugated to FITC; C5b (DAKO, Carpentaria, CA) and C4d (QUIDEL, San Diego, CA) were unconjugated ab detected by Alexa Fluor® 488 goat anti-mouse secondary antibody (Abcam, Cambridge, MA) to assess antibody binding and complement activation in the graft. Paraffin-embedded tissues were sectioned, stained using hematoxylin and eosin (H&E) and Periodic acid-Schiff, and examined by an experienced pathologist. Paraffin-embedded tissues were used for immunohistochemistry (IHC) of human adenovirus. Human adenovirus unconjugated Ab (Abcam, Cambridge, MA) was detected by HRP horse Anti-mouse secondary antibody (vectorlabs, California, US) to assess human adenovirus infiltration in the graft.

### Reporting summary
Further information on research design is available in the Nature Portfolio Reporting Summary linked to this article.

## Data availability
All data supporting this study are available within the article and its related supplementary information file. All raw data for graphs in this study are included in a single excel file labeled "source data," and provided with this paper. All microscopic images are available from the corresponding author on request. Source data are provided with this paper.

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

## Author contributions

D.E. and K.Y. designed, supervised, and conducted the experiments, as well as interpreted data and wrote the manuscript. Y.H. conducted the experiments, analyzed data, and contributed to writing of the manuscript. M.S. conducted in vitro experiments, analyzed data, and assisted with animal care and surgical procedures. H.I. contributed to animal care, surgical procedures, and the writing of the manuscript. W.C. and A.Sh. performed histologic analyses. A.Sc. contributed to animal care, surgical procedures, and the writing of the manuscript. D.G. conducted in vitro experiments. A.A. and A.Z. contributed to animal care, data analysis, and editing of the manuscript. K.Ko. contributed to day-to-day animal care as well as anesthesia for procedures. K.Ku., F.R. and L.S. contributed to data analysis and figure preparation. L.B. assisted with procurement and preservation procedures. K.D., L.P, M.L. and D.A. contributed to experimental design, interpretation of data, review of the manuscript, and D.A. provided the genetically engineered pigs. D.W. and A.C. contributed to experimental design and interpretation of data.

## Competing interests

This study was supported by United Therapeutics Corporation. K.K. (Kuravi), F.R., L.S., L.B. and D.A. are employees of Revivicor, a subsidiary of United Therapeutics Corporation. K.D., L.P., and M.L. are employees of United Therapeutics Corporation. The remaining authors declare no other competing interests.
