## [Peer Review File · Nature Communications]

Consistent survival in consecutive cases of life-supporting porcine kidney xenotransplantation using 10GE source pigsREVIEWER COMMENTS

Reviewer #1 (Remarks to the Author):

General Comment

The authors present long-term survival data for pig-to-NHP kidney xenotransplantation with conditions mimicking future pig-to-human kidney xenotransplantation. Porcine kidney grafts had >3 hours of cold ischemia time prior to transplantation and FDA-approved immunosuppression was used, specifically avoiding CD40/CD154 costimulatory blockade.

These data are important in the next step of safety/feasibility prior to clinical trials involving xenotransplantation.

Major comments:

- (1) The authors aim to use only FDA approved and clinically available immunosuppression; however, a recombinant primate derivative of an anti-CD40 monoclonal antibody (2C10R) forms the back-bone maintenance immunosuppression in 5 of the 6 baboon recipients. While 2C10 is similar to 2C10R (ref 1), it would strengthen the paper to clarify the distinction between 2C10R and CD40/CD154 costimulatory blockade.
- (2) How were the pig-kidneys transplanted, 2 kidneys en-bloc or 1 kidney attached to femoral or iliac vessels?

Minor comments:

- (a) The authors reference their own work that is not-yet published. Please insert published references where “manuscript submitted” is mentioned, if not available, then remove.
- (b) It would be helpful to include a link to online methods in the reference section. This can be used in place of “(see online methods)” in the text.
- (c) What was the immunosuppressive regimen for the one baboon with hyperacute rejection?
- (d) For figure 2, panel b: please change the y-axis scale to have a maximum of 5 mg/dL. The outcome of kidney failure defined as > 10 mg/dL can be mentioned in the figure legend, if needed.
- (e) Are there any physiologic data available like urine output, blood pressure, serum

electrolyte levels?

(f) Please provide the size of the baboons (weight) and the pig-kidney grafts (weight) in the main document, as opposed to the supplementary data.

(g) Was the specific cold ischemia time for each of the 6 xenotransplants reported?

(h) In the discussion, reference to a website can be included in the references, as opposed to including the entire hyperlink in the text.

Reference:

1) Michaels AJ, Stoppato M, Flores WJ, Reimann KA, Engelman KD. Anti-CD40 antibody 2C10 binds to a conformational epitope at the CD40-CD154 interface that is conserved among primate species. *Am J Transplant*. 2020 Jan;20(1):298-305. doi: 10.1111/ajt.15574. Epub 2019 Sep 12. PMID: 31430418; PMCID: PMC6940519.

Reviewer #2 (Remarks to the Author):

The potential for xenotransplantation to provide a limitless supply of organs for transplantation has long been appreciated, but the recent studies in preclinical models, recently deceased humans and with two porcine heart transplants under compassionate use exemption, the field is in the midst of concerted efforts toward bringing about clinical trials. It is in this context that results presented attempting to prevent xenograft rejection with clinically available agents is of interest. There are a number of weaknesses in the work that lessen its impact.

1. The introductory sentence seems overly optimistic claiming success in clinical studies when both human recipients of cardiac xenografts were lost in <2 months.

2. The primary claim in the current work is that kidneys from 10GE donors can succeed with conventional (ie FDA approved) immunosuppression. Unfortunately, the reliance on a primary end point of only six months is seem inadequate to claim success. Longer term outcome of at least a year is needed to understand the full efficacy of the regimen and to be comparable to other major publications in the field. Also, that the authors state (line 226) that "... rejection often occurs after 90 days" is discordant with their use of a 180-day MST

end point for the trial overall. Thus, authors are obligated to show the full survival data to at least 1 year and preferably beyond as reported in the Anand paper they reference. The need for this degree of follow-up is essential as the longest survivors in their series appear to show a gradually rising creatinine beyond the 180-day time point. In addition, it appears that not all animals have reached the 180-day endpoint; that the mean survival of the group was >180 days is not sufficient. As a result of these weaknesses, their conclusory assertion that the results have “immediate implications for imminent xenotransplantation trial design” (line 234) are unfounded. The authors should provide at least 1 year survival data, ideally for all animals in the cohort.

3. In addition, the emphasis on use of clinically approved reagents seems short sighted given diverse data indicating that targeting CD40 or CD40L in xenotransplantation is powerful adjunct in control of the xenoreponse and that such reagents are progressing rapidly through clinical trials. Studies comparing conventional immunosuppression versus anti-CD40/40L targeting have shown superiority to those with conventional approved agents. Furthermore, one such reagent was applied in the most recent pig to human cardiac transplant, presumably under a compassionate use exemption. Since the models used by different groups vary, for example the current using baboons as recipients while the most recent analysis by Anand et al, relied on cynos, the current analysis would have been stronger if a head-to-head comparison of bela based immunosuppression costimulation blockade was compared to antiCD40/40L, all in a baboon xenotransplant model.

4. The authors note that current FDA regulations regarding recovery of genetically modified porcine organs in a DPF facility will incur an obligatory period of cold ischemia. However, their statement (line 198) that “no prior studies have achieved long term xenograft survival with any cold ischemia time” requires corroboration or referencing (it is unclear how would the authors know this to be true). And, in general, all organ recoveries incur some CIT unless using a totally ischemia free approach. They authors should be more precise in their statements.

5. The authors note use of the Yamada Lab screening methodology as relevant to the outcome of recipients in the study. Given the important of this statement to the presented

work, the authors should share the specifics of that screening protocol so that readers can assess its impact on study outcomes and determine if it differs substantively from those already in use. The authors might also share how the multi-gene insertion approach used in their work differs from the work of Anand et al, whose work they reference in the discussion.

6. The use of oxygen preservation is of interest but was not tested in a controlled manner to permit understanding of its impact in the current results.

7. The work would be strengthened by documenting function of the inserted genes rather than merely expression.

Reviewer #1 (Remarks to the Author):

General Comment

The authors present long-term survival data for pig-to-NHP kidney xenotransplantation with conditions mimicking future pig-to-human kidney xenotransplantation. Porcine kidney grafts had >3 hours of cold ischemia time prior to transplantation and FDA-approved immunosuppression was used, specifically avoiding CD40/CD154 costimulatory blockade.

These data are important in the next step of safety/feasibility prior to clinical trials involving xenotransplantation.

RESPONSE: we appreciate the reviewer's recognition of the value of this safety/feasibility study as we approach clinical trials in xenotransplantation.

Major comments:

- (1) The authors aim to use only FDA approved and clinically available immunosuppression; however, a recombinant primate derivative of an anti-CD40 monoclonal antibody (2C10R) forms the back-bone maintenance immunosuppression in 5 of the 6 baboon recipients. While 2C10 is similar to 2C10R (ref 1), it would strengthen the paper to clarify the distinction between 2C10R and CD40/CD154 costimulatory blockade.

- **RESPONSE: thank you for this comment and for the opportunity to make a correction and clarification in our report. We used 2C10R4, a mouse-rhesus IgG4 chimeric antibody which has been well-described in the xenotransplantation literature – the referenced article (Michaels et al, Am J Transplant, 2020) cites pioneering research by Mohiuddin et al, published in Nature Communications in 2016, as well as studies by other groups (Langin, Nature, 2018) (Iwase, Transpl Immunol, 2015) (Iwase, Xenotransplantation, 2015). We agree that it is important to clarify the mechanism of 2C10R4 as it relates to CD40/CD154 blockade, as other anti-CD40 agents have been used for other purposes, and we have added language clarifying our choice of agent in the methods section, as well as additional language describing the targets of each of the other anti-CD40 (KPL-404) or anti-CD154 (AT-1501) agents mentioned.**

➤ REVISION:

1. Lines 124-126: “The CD40/CD154 costimulation blockade-based regimen included the mouse-rhesus IgG4 chimeric anti-CD40 monoclonal antibody (2C10R4, targeting the 2C10 epitope of the CD40 molecule, dosed at 25 mg/kg twice weekly after loading dose)”
2. Lines 251-252: “The University of Maryland included CD40/CD154 blockade, KPL-404 (Kiniksa Pharmaceuticals), which targets the 2C10 binding epitope on CD40 (Kiniksa Pharmaceuticals), and tegoprubart (Eledon Pharmaceuticals), which targets CD154 (CD40L), for their two clinical pig-to-human heart transplants, respectively.”

(2) How were the pig-kidneys transplanted, 2 kidneys en-bloc or 1 kidney attached to femoral or iliac vessels?

- **RESPONSE: pig kidneys were transplanted orthotopically in the retroperitoneum, with graft renal vessels anastomosed to the aorta and IVC. This critical point has been clarified in the methods section.**

➤ **REVISION: Lines 142-143, “Each baboon recipient received one porcine kidney, transplanted orthotopically with porcine renal vein and artery anastomosed to the inferior vena cava and the aorta, respectively.”**

Minor comments:

(a) The authors reference their own work that is not-yet published. Please insert published references where “manuscript submitted” is mentioned, if not available, then remove.

- **RESPONSE: we have removed references to unpublished work.**

(b) It would be helpful to include a link to online methods in the reference section. This can be used in place of “(see online methods)” in the text.

- **RESPONSE: we agree that this would facilitate interpretation of our results and improve readability of the article. We will work with the editor to include this link in the accepted article.**

(c) What was the immunosuppressive regimen for the one baboon with hyperacute rejection?

- **RESPONSE: the baboon who hyperacutely rejected the graft received an anti-CD40-based regimen (2C10R4). We added a sentence to clearly state this animal’s immunosuppression regimen in the results section, as we agree that this is important information to clarify.**

➤ **REVISION:**

1. **Lines 145-146, “One recipient (Group 2, CD40/CD154 costimulatory blockade maintenance immunosuppression)...”**
2. **Table 1: we also revised the paper to include a table summarizing these key details.**

(d) For figure 2, panel b: please change the y-axis scale to have a maximum of 5 mg/dL. The outcome of kidney failure defined as > 10 mg/dL can be mentioned in the figure legend, if needed.

- **RESPONSE:** we based this y-axis on our primary endpoint according to our ACUC protocol of a creatinine of 10mg/dL. Other recent publications, including the recent paper published by Anand et al in Nature (Anand, Nature, 2023), used a serum creatinine y-axis maximum of 12mg/dL. It is useful to maintain the current scale (maximum of 10mg/dL) to facilitate comparison to this publication as well as our own previous publications (Takeuchi, Xenotransplantation, 2021).

(e) Are there any physiologic data available like urine output, blood pressure, serum electrolyte levels?

- **RESPONSE:** thank you for this comment. We have revised the article to include a panel of serum electrolytes (Sodium, Potassium, Chloride, and Calcium), and included graphs with these data in the Supplementary Information section. We also agree that additional parameters (including blood pressure and other electrolytes) are critical to safe and successful clinical translation of xenotransplantation. We are actively collecting this data now, and additional studies will include these parameters.

➤ **REVISION:**

1. **Lines 150-154:** “The xenograft kidneys have also maintained normal concentrations of sodium, potassium, and chloride. Mild-to-moderate hypercalcemia was observed in long-term survivors (see supplementary information), a physiologic change that has been previously reported after porcine renal xenograft transplantation in a pig-to-macaque model”
2. **Supplementary figure 2: Additional physiologic data: Serum Electrolytes.** Graphs for each electrolyte (Na, K, Cl, and Ca) handling were created and include information from each recipient.

(f) Please provide the size of the baboons (weight) and the pig-kidney grafts (weight) in the main document, as opposed to the supplementary data.

- **RESPONSE:** we agree that this is important information and we revised the main text to include weight ranges for source pigs (17, 18, and 24kg) as well as baboon recipients (8.5, 9.5, 10, 10, 10.5, and 12.5kg). We also included size of pig kidney. Given that source pig was growth hormone receptor knockout (GHRKO), 10GE kidneys were smaller than expected relative to GalTKO without GHRKO pigs of similar body weight: size of pig kidneys used in this study was similar to size of pig kidneys used in previous studies using GalTKO without GHRKO source pigs (Tanabe and Yamada et al, Am J Transplant, 2017).

➤ **REVISION:**

1. **Line 115-116: “The baboons ranged from 2-4 years old and weighed between 8 to 12.5 kg. Equal numbers of male (3) and female (3) baboons were included.”**
2. **Lines 129-130: “The porcine kidneys were procured from three 10GE source pigs, ranging from 17 to 24kg in weight at the time of organ procurement”**

(g) Was the specific cold ischemia time for each of the 6 xenotransplants reported?

- **RESPONSE: yes, specific cold ischemic time for each of the 6 cases was recorded and we have revised the main text to include a range (3-5 hours) of ischemic time, as well as one additional table which summarizes recipient weights, source pig weights, preservation modality, and preservation time.**

➤ **REVISION:**

1. **Table 1: we revised the paper to include a table summarizing these key details (reproduced below).**

Recipient ID	Recipient Sex	Recipient Weight (kg)	Pig Weight (kg)	Xenokidney Weight (g)	Preservation Modality	Preservation Time (hours)
B001320	M	9.5	17	-	Lifeport	3.5 hours
B003520	M	8.5	17	-	XVIVO	3.0 hours
B000820	F	10.5	24	56.4	XVIVO	5.2 hours
B004519	F	10	24	51.2	Lifeport	4.0 hours
B009819	M	10.5	18	38.6	Lifeport	3.7 hours
B010419	F	12.5	18	38.6	Lifeport	4.5 hours

(h) In the discussion, reference to a website can be included in the references, as opposed to including the entire hyperlink in the text.

- **RESPONSE: thank you for this suggestion. We have moved this to the reference section.**

Reviewer #2 (Remarks to the Author):

The potential for xenotransplantation to provide a limitless supply of organs for transplantation has long been appreciated, but the recent studies in preclinical models, recently deceased humans and with two porcine heart transplants under compassionate use exemption, the field is in the midst of concerted efforts toward bringing about clinical trials. It is in this context that results presented attempting to prevent xenograft rejection with clinically available agents is of interest. There are a number of weaknesses in the work that lessen its impact.

1. The introductory sentence seems overly optimistic claiming success in clinical studies when both human recipients of cardiac xenografts were lost in <2 months.

- **RESPONSE: we agree that an appropriate introductory sentence needs to highlight that the decedent studies and the two clinical cases represent “short-term” successes, in contrast to the long-term survivals described in pig-to-NHP studies. We have revised the text to reflect this distinction.**

➤ **REVISION: lines 38-40, now reads “Xenotransplantation represents a possible solution....and **short-term** success in recent human decedent and clinical studies.”**

2. The primary claim in the current work is that kidneys from 10GE donors can succeed with conventional (ie FDA approved) immunosuppression. Unfortunately, the reliance on a primary end point of only six months is seem inadequate to claim success. Longer term outcome of at least a year is needed to understand the full efficacy of the regimen and to be comparable to other major publications in the field. Also, that the authors state (line 226) that “... rejection often occurs after 90 days” is discordant with their use of a 180-day MST end point for the trial overall. Thus, authors are obligated to show the full survival data to at least 1 year and preferably beyond as reported in the Anand paper they reference. The need for this degree of follow-up is essential as the longest survivors in their series appear to show a gradually rising creatinine beyond the 180-day time point. In addition, it appears that not all animals have reached the 180-day endpoint; that the mean survival of the group was >180 days is not sufficient. As a result of these weaknesses, their conclusory assertion that the results have “immediate implications for imminent xenotransplantation trial design” (line 234) are unfounded. The authors should provide at least 1 year survival data, ideally for all animals in the cohort.

- **RESPONSE: we appreciate the reviewer’s comments and reasoning regarding long term follow-up, and we agree that longer term follow-up (beyond six months) is critical prior to initiation of clinical trial. We are actively following recipients in this cohort (beyond six months) to be included in eventual IND submission. Still, we believe these data are important to publish at this time point for the following reasons:**

- **180 days is an important milestone: while our official nonclinical study will follow recipients out to one year, our conversations with the FDA have suggested to us that six months remained an important checkpoint for evaluation of results of this study.**

- **Immunologic differences in between baboon and macaque recipient models complicate direct comparisons of long-term survival after xenotransplantation. The underlying mechanisms are not well-understood, but this phenomenon has been described in islet xenotransplantation (Hawthorne, *Front Immunol*, 2022) and the differences are pronounced in renal xenotransplantation. For instance, while Anand et al recently published two-year survival of renal xenografts in a pig-to-macaque model, the longest reported xenograft survival in a pig-to-baboon to date is 260 days (Yamamoto, *Transplantation*, 2019). Two recipients in this study have xenograft survivals of 265 days at this time point, which would make them the longest published xenograft survivals in a pig-to-baboon renal xenotransplantation model.**
 - **However, the most salient observation among these data is not that we have isolated longest-term survivors: the emphasis of this paper is the consistency of these results in consecutive transplanted cases. In the recent study published by Anand et al, 3/8 of recipients experienced graft loss in the first 30 days, as compared to only 1/6 recipients in this study.**

- **To the reviewer's last point – 180-day survival with CNI-based, conventional immunosuppression is not sufficient to impact xenotransplantation clinical trial design – we agree that more data are needed to replicate these results, but we do believe that these data should prompt careful reconsideration of immunosuppression protocols prior to any additional clinical cases for the following reasons:**
 - **The importance of CD40/CD40L blockade in xenotransplantation is based on decades of landmark studies, but dogma around CD40/CD40L blockade runs the risk of obscuring the heterogeneity among these agents. As the other reviewer observed, the target epitope and the specific reagent is important, and not all anti-CD40/CD40L agents are effective. For instance, the backbone of immunosuppression for the second EIND pig-to-human heart transplant at the University of Maryland was an anti-CD40L mAb called tegoprubart (Eledon Pharma) which has not demonstrated long-term survival in a single published pig-to-NHP study, and which is not yet clinically approved for use in allogeneic transplantation.**
 - **In order to determine whether CD40/CD40L blockade is necessary for xenotransplantation, control studies must demonstrate that CNI-based immunosuppression regimens are subject to the same clinical standards, with careful monitoring to ensure therapeutic**

drug levels. In our review of other studies using conventional immunosuppression, we were not able to confirm that recipients were maintaining therapeutic drug levels. Our single case (and first attempt) of long-term and ongoing survival using CNI-based immunosuppression titrated to therapeutic level calls into question the conclusions of previous studies which have been used to assert that CD40/CD40L blockade was necessary.

3. In addition, the emphasis on use of clinically approved reagents seems short sighted given diverse data indicating that targeting CD40 or CD40L in xenotransplantation is powerful adjunct in control of the xenoresponse and that such reagents are progressing rapidly through clinical trials. Studies comparing conventional immunosuppression versus anti-CD40/40L targeting have shown superiority to those with conventional approved agents. Furthermore, one such reagent was applied in the most recent pig to human cardiac transplant, presumably under a compassionate use exemption. Since the models used by different groups vary, for example the current using baboons as recipients while the most recent analysis by Anand et al, relied on cynos, the current analysis would have been stronger if a head-to-head comparison of bela based immunosuppression costimulation blockade was compared to antiCD40/40L, all in a baboon xenotransplant model.

- ***RESPONSE: we acknowledge and appreciate the decades of research from multiple investigative groups that have demonstrated long-term survival of xenografts (kidney, heart, and islet) with reagents targeting CD40 or CD40L and we agree that head-to-head comparison of conventional vs CD40/CD40L blockade-based immunosuppression is critical. We are undertaking this direct comparison study now.***
 - ***We also agree that some reagents targeting CD40 and CD40L have demonstrated promising early results in clinical trials of allotransplantation and we have revised the text to reflect this. However, we were concerned that one such agent, iscalimab, failed to demonstrate non-inferiority in clinical trials, and we are not aware of any anti-CD40 or anti-CD40L reagent which has demonstrated superiority in clinical trials of allotransplantation.***
 - ***Although there have been previous studies comparing conventional immunosuppression to CD40/CD40L blockade-based immunosuppression in solid organ xenotransplantation, we are concerned (as mentioned above) that recipients in the conventional immunosuppression group were not maintaining therapeutic drug levels.***
 - ***Moreover, as mentioned above, these reagents are heterogeneous and efficacy varies depending the specific reagent used. AT-1501 (tegoprubart) was used in the recent University of Maryland case even though there were no published studies (to our knowledge) demonstrating long-term solid organ xenograft survival with use of this specific reagent.***

4. The authors note that current FDA regulations regarding recovery of genetically modified porcine organs in a DPF facility will incur an obligatory period of cold ischemia. However, their statement (line 198) that “no prior studies have achieved long term xenograft survival with any cold ischemia time” requires corroboration or referencing (it is unclear how would the authors know this to be true). And, in general, all organ recoveries incur some CIT unless using a totally ischemia free approach. They authors should be more precise in their statements.

- **RESPONSE: thank you for the opportunity to clarify this point. We have revised this statement accordingly.**

- **REVISION: Lines 224-225, “To our knowledge, no prior studies have described long-term xenograft survival after prolonged (>3 hours) cold ischemic time...”**

5. The authors note use of the Yamada Lab screening methodology as relevant to the outcome of recipients in the study. Given the important of this statement to the presented work, the authors should share the specifics of that screening protocol so that readers can assess its impact on study outcomes and determine if it differs substantively that those already in use. The authors might also share how the multi-gene insertion approach used in their work differs from the work of Anand et al, whose work they reference in the discussion.

- **RESPONSE: we appreciate this comment and we agree that details of the screening methodology should be included in the text as they are relevant to the outcome of this study. We have revised the paper to include these details (as well as a flowchart summarizing our recipient selection paradigm) as supplementary information. A forthcoming publication (recently accepted) detailing this screening methodology will also be referenced here.**

- **REVISION:**

1. **Lines 109-114: “Prospective recipients were evaluated for anti-pig antibodies by complement dependent cytotoxicity (CDC) using peripheral blood mononuclear cells (PBMCs) derived from triple antigen knockout source pigs. The presence of donor-specific antibodies was assessed by IgG and IgM antibody binding of baboon sera to 10GE-derived PBMCs. Recipients were selected according to screening methodology developed by The Yamada Lab at Johns Hopkins University.”**
2. **Supplementary Figure 1: Flowchart outlining screening and selection process for prospective recipients of 10GE organs.**

6. The use of oxygen preservation is of interest but was not tested in a controlled manner to permit understanding of its impact in the current results.

- **RESPONSE:** we agree that additional studies should evaluate the role of oxygenated vs non-oxygenated preservation modalities in xenotransplantation. Although our current study was not designed to evaluate the impact of oxygenation during preservation, we have revised the paper to include a table which summarizes recipients by preservation modality and transportation / cold ischemic time.

➤ **REVISION:** Table 1 (referenced above)

7. The work would be strengthened by documenting function of the inserted genes rather than merely expression.

- **RESPONSE:** we agree that evaluation of the function of inserted genes is critical prior to proceeding with clinical trials. We appreciate the suggestion and will undertake this analysis to include in subsequent studies.

REVIEWER COMMENTS

Reviewer #1 (Remarks to the Author):

The authors have gone a long way in addressing this reviewer's concerns. Just a few minor suggestions:

- 1) A hyperlink remains in the discussion (line 229)
- 2) I understand the authors wanting to keep the y-axis scale at a maximum of 12 mg/dL. However, the monkeys are small in weight with low muscle mass and daily creatinine production will be low. By compressing the serum creatinine data in the figure, it comes across as hiding information. If the y-axis is kept at 12 mg/dL maximum, then a zoomed in insert should be added to the figure.
- 3) Thank you for adding the electrolyte data. Again, the y-axis scale on the figures is problematic, however. From a physiology standpoint, serum sodium levels below 100 or above 160 mMol/L are not compatible with life. Why include SNa ranges up to 200 and down to 0 mMol/L on the y-axis range? This again compresses the data. While the other electrolyte y-axis scales could be adjusted, this is particularly important for the serum Calcium levels, which were higher than expected.

Reviewer #2 (Remarks to the Author):

Thank you to the authors for their efforts at revision and for the opportunity to review changes.

There remain a number of issues that the authors should address:

- 1) If the data are available, please provide size of the kidney pre and post -transplant.
- 2) The authors refer to long term survival to describe their results. I still do not think 6-month mean or median qualifies as long-term survival and remain stuck on the issue of duration of follow-up. This is important in that the work tries to claim that kidneys from 10GE donors succeed with conventional, FDA approved, immunosuppression. The issue remains that 6-month survival is insufficient for the claims made in my opinion. It is unlikely to be adequate for the FDA for a clinical trial and 6-month survival would not meet a clinical standard of success for most patients, I suspect. This, plus the fact that there are other examples of xenograft survival for much longer by other teams, makes the impact of the

work rest on use of co-stimulation blockade of B7 versus co-stimulation blockade of CD40/CD40L. Expecting that CD40/CD40L agents will likely be approved based on early data, lessens the novelty of belatacept use. The work would better suited for a premier journal with a full year of follow-up. The statement on page 5 lines 77-78 that claim “long-term survival in five of 6 recipients...” while encouraging is not convincing.

3) Figure 2 survival data seems not to have been updated from the original submission making one wonder if there have been graft losses in the intervening period between submissions. Please send the updated data.

4) Similarly, the authors claim the first >3 hour cold time successful transplants but provide no evidence that this is beneficial. The claim should be deemphasized.

Response to Reviewers, Resubmission #2

Reviewer #1 (Remarks to the Author):

The authors have gone a long way in addressing this reviewer's concerns. Just a few minor suggestions:

- 1) A hyperlink remains in the discussion (line 229)
 - **Response: we have deleted this hyperlink.**

- 2) I understand the authors wanting to keep the y-axis scale at a maximum of 12 mg/dL. However, the monkeys are small in weight with low muscle mass and daily creatinine production will be low. By compressing the serum creatinine data in the figure, it comes across as hiding information. If the y-axis is kept at 12 mg/dL maximum, then a zoomed in insert should be added to the figure.
 - **Response: we understand this reviewer's concern. Given the inclusion of new data with two recipients who developed renal failure (with elevated creatine >8) we believe it is important to maintain a maximum of 12mg/dL in the main figure.**

- 3) Thank you for adding the electrolyte data. Again, the y-axis scale on the figures is problematic, however. From a physiology standpoint, serum sodium levels below 100 or above 160 mMol/L are not compatible with life. Why include SNa ranges up to 200 and down to 0 mMol/L on the y-axis range? This again compresses the data. While the other electrolyte y-axis scales could be adjusted, this is particularly important for the serum Calcium levels, which were higher than expected.
 - **Response: we appreciate the reviewer's observation and we have adjusted the y-axes of each electrolyte graph (Na, Cl, and Ca) to allow for closer inspection of the data.**

Reviewer #2 (Remarks to the Author):

Thank you to the authors for their efforts at revision and for the opportunity to review changes. There remain a number of issues that the authors should address:

- 1) If the data are available, please provide size of the kidney pre and post -transplant.
 - **Response: we are grateful to the reviewer for the opportunity to include these data, which we believe is important in the context of the broader debate around graft growth and whether the use of miniature pigs or growth hormone receptor KO pigs is necessary for clinical xenotransplantation. In contrast to our previous report using kidneys from GalTKO source pigs, we found minimal graft growth using kidneys derived from 10GE source pigs which are GHRKO in our pig-to-baboon model. We**

have added a graph describing graft growth as measured by ultrasound at regular intervals in Supplementary Information (Supplementary Figure 5).

2) The authors refer to long term survival to describe their results. I still do not think 6-month mean or median qualifies as long-term survival and remain stuck on the issue of duration of follow-up. This is important in that the work tries to claim that kidneys from 10GE donors succeed with conventional, FDA approved, immunosuppression. The issue remains that 6-month survival is insufficient for the claims made in my opinion. It is unlikely to be adequate for the FDA for a clinical trial and 6-month survival would not meet a clinical standard of success for most patients, I suspect. This, plus the fact that there are other examples of xenograft survival for much longer by other teams, makes the impact of the work rest on use of co-stimulation blockade of B7 versus co-stimulation blockade of CD40/CD40L. Expecting that CD40/CD40L agents will likely be approved based on early data, lessens the novelty of bela use. The work would better suited for a premier journal with a full year of follow-up. The statement on page 5 lines 77-78 that claim “long-term survival in five of 6 recipients...” while encouraging is not convincing.

- **Response:** we acknowledge the reviewer’s concerns and we have modified terminology throughout (see point “d”) the paper to moderate our description of the duration of survival. We would like to clarify the following points.

a) We agree that continued follow-up is important and we have extended the observation period to nine months. However, the emphasis of this paper

is not on isolated long-term survivals; rather these data demonstrate consistency of results that have never before been achieved.

- In previous studies (including the recent publication by Anand et al), a significant proportion of recipients reject their grafts within the first month (>40% in the study described by Anand et al, and perhaps higher considering that two recipients underwent delayed native nephrectomy on POD20, rather than at the time of transplantation). In contrast, our results demonstrate no graft loss within the first month among low-risk recipients.
- Consistency of results is important to the FDA and is critical to successful clinical translation of these results. While there may not yet be consensus regarding what duration of graft survival constitutes clinical success, 40% graft loss within the first month would undoubtedly not meet a clinical standard of success for most patients.
- Moreover, it is worth clarifying that there is no published guidance from the FDA establishing what duration of survival in NHP studies will be required to initiate clinical studies. However, guidance from earlier conversations with regulatory bodies has supported the view described by Cooper and colleagues: “the national regulatory authorities are likely to require demonstration of...survival of a life-supporting pig kidney in a NHP for six months or longer in a series of at least six consecutive experiments...as maintaining immunosuppressed NHPs consistently for long periods is much more difficult than maintaining a patient in a hospital setting” (Cooper, EBioMedicine, 2021).

b) This model employs a pig-to-baboon rather than pig-to-macaque transplantation model, which we (among others) believe to be a more immunostriking model for xenotransplantation. With the inclusion of extended observation data up to nine months post-transplantation, three of the baboons in this study would be the longest survivors of pig-to-baboon kidney xenotransplantation ever published.

- Moreover, inclusion of extended survival data (up to nine months observation in consecutive cases) brings the mean survival to 205 days and the median survival to 261 days, which, to our knowledge, would be the highest mean and median survival in a case series of

6 months check point (10GE KTx)

9 months check point (10GE KTx)

consecutive solid organ xenotransplants ever published, in either macaques or baboons.

- As this is part of a larger study, longer term survival data (to one year or longer) will lead to marked delays in publication.

c) This study does not claim that conventional immunosuppression is superior to CD40/CD40L costimulatory blockade; rather this study seeks to challenge the longstanding dogma that costimulatory blockade is necessary. In the preceding decades, the longest survivor of pig-to-NHP solid organ xenotransplantation without CD40 costimulatory blockade was 32 days (Yamamoto, Transplantation, 2019); in our study, we have included survival that is nine times longer (currently 272 days) using conventional immunosuppression. Given that this survival is longer than any previously published pig-to-baboon kidney survival (regardless of immunosuppression regimen) we believe it is worthy of publication at this time. Of course, this regimen will need to be evaluated in subsequent head-to-head comparison studies, which we are undertaking, but these data are timely and help to bridge a critical division between decades of NHP studies and recent high-profile human decedent studies which have not used CD40 costimulatory blockade.

d) We acknowledge the reviewer's concern with our description of six-month survival as "long-term." Accordingly, we have changed the title to omit this term and to underscore the consistency of our results, and we have moderated this term throughout the study.

- The title now reads "Consistent survival in life-supporting porcine kidney xenotransplantation: initial experience using 10GE source pigs with nine-months follow-up."
- The statement on page 5 Line 77-78 has been qualified to emphasize the consistency of results: "...we present the first demonstration of consistent survival in consecutive cases of pig-to-NHP kidney xenograft transplantation."

3) Figure 2 survival data seems not to have been updated from the original submission making one wonder if there have been graft losses in the intervening period between submissions. Please send the updated data.

- **Response:** two grafts were lost recently and three grafts remain. As described above, we have included updated survival data in this resubmission.

4) Similarly, the authors claim the first >3 hour cold time successful transplants but provide no evidence that this is beneficial. The claim should be deemphasized.

- **Response:** we would like to clarify that we are not making the claim that preservation time is beneficial in porcine kidney xenotransplantation; rather, we believe that cold ischemic time will be unavoidable in clinical porcine kidney xenotransplantation given the regulatory requirements around animal housing and procurement. Accordingly, we believe it is important to include and to describe clinically relevant preservation times and strategies in feasibility studies of pig-to-NHP kidney xenotransplantation. To our knowledge, our study includes the first description of >6 months survival after >3 hrs of cold preservation time.

REVIEWER COMMENTS

Reviewer #2 (Remarks to the Author):

Since the last submission a graft loss is noted, likely rejection related but not specified (histology should be provided), and two of the three remaining recipients have rising creatinine levels (apparently no biopsies performed). The 180day - 365 day window is likely critical as noted by the rising creatinine levels. As per my prior critiques, the authors should have a year of follow-up on the remaining three cases to have a meaningful report, and at that time, I would support publication whether or not there are additional graft losses.

REVIEWER COMMENTS

Reviewer #2 (Remarks to the Author):

Since the last submission a graft loss is noted, likely rejection related but not specified (histology should be provided), and two of the three remaining recipients have rising creatinine levels (apparently no biopsies performed). The 180day - 365 day window is likely critical as noted by the rising creatinine levels. As per my prior critiques, the authors should have a year of follow-up on the remaining three cases to have a meaningful report, and at that time, I would support publication whether or not there are additional graft losses.

We appreciate your comments and we have provided the full follow-up period for all recipients in our study: the remaining three recipients were euthanized on POD 337, POD 278, and POD 285. We have included complete histology in this resubmission. We have updated the results section as follows to include these data:

- “All recipients with low levels of preformed anti-source pig Nabs accepted their grafts and maintained stable graft function for 337, 285, 278, 252, and 165 days (Figure 2a and 2b, details in figure legend). Among all recipients, mean and median survival was 220 days and 261 days respectively; among five low-risk recipients, mean survival was 263 days (± 49.5 [CI: 213 – 313 days]) and median was 278 days. Four of five low-risk recipients maintained graft function beyond six months (histology of xenograft biopsies at this timepoint shown in Figure 3); three of the remaining four recipients maintained graft function beyond nine months. One recipient (B3520, Group 2, CD40/CD154 costimulatory blockade maintenance immunosuppression) developed antibody-mediated rejection at the time of necropsy on POD278 (Figure 4), with IgG and IgM antibody-binding as well as a modest increase in circulating donor specific antibodies (DSA) (See supplementary information). Another recipient (B0820, Group 1, CNI-based immunosuppression) developed progressive renal failure which met endpoint criteria on POD285 and was found on necropsy to have histologic evidence of acute cellular xenograft rejection with lymphocyte infiltrate. **The other three recipients (B9819, B4519, and B1320, all Group 2, CD40/CD154 costimulatory blockade maintenance immunosuppression) lost their grafts suddenly on POD165, POD252, and POD337 respectively. In each case, rapid decline in graft function was preceded by symptoms of upper respiratory tract infection and adenovirus seroconversion consistent with new infection. Necropsy kidney specimens stained positive for adenovirus (Figure 5). One of these recipients (B4519, necropsy histology included in Figure 4) also had weak IgG and IgM binding (Figure 5), although elicited DSA was negative (see supplementary information). Among the remaining two recipients, there were no elicited DSAs, and only minimal and nonspecific IgG/IgM binding assessed by immunofluorescence of necropsy xenograft specimens (Figure 5).”**

REVIEWERS' COMMENTS

Reviewer #2 (Remarks to the Author):

The additional data accrued over the past 3-6 months helps to paint a realistic picture of outcomes. That all grafts were lost suggests that the regimen is likely not superior to others already reported and that it was perhaps over immunosuppressive given the suggestion that three were lost to viral infection or a combination of viral infection and rejection. These are important findings that help to define where this work fits among other in the literature. The authors are to be congratulated on the successful completion of these most challenging studies.